# Clinical manifestations of bed bug bites: A systematic review of case reports

**Julian Felipe Porras-Villamil** *, **Zachary C. DeVries**

Department of Entomology, University of Kentucky, Lexington, Kentucky, United States of America

* julian.porras@uky.edu

## Abstract

### Importance

Clinical characterization of bed bug (*Cimex* spp.) bite manifestations remains limited.

### Objective

This systematic review synthesizes case reports to describe dermatological, systemic, and psychological outcomes of bed bug bites.

### Evidence review

Following PRISMA guidelines, multiple databases, including Pubmed and Scopus, were searched using terms such as "bed bug," "*Cimex lectularius*," "*Cimex hemipterus*," "case report," and "case series." Extracted data included demographics, species, lesion type, anatomical distribution, and clinical manifestations. Quantitative and qualitative analyses were conducted.

### Results

Eighty-four patients were included: 76 with *C. lectularius* (90.5%) and, 8 with *C. hemipterus* (9.5%). Sex distribution was balanced (47.6% females, 52.4% males), and mean ages were similar (females 37.5±22.1, males 42.3±21.27 years; p=0.332). Extremities were commonly affected, arms were reported in 71.4% of cases, legs in 59.5% and trunk in 34.5%. Psychological symptoms were registered in a handful of case reports (Anxiety 9.5%, Stress 9.5%, Hopelessness 9.5%). Systemic symptoms, generally associated with anemia, were reported in cases related to severe infestations in neglected patients. Some differences in systemic and local signs and symptoms were detected depending on age group.

provided the original author and source are credited.

**Data availability statement:** All relevant data are within the paper and its Supporting information files.

**Funding:** This work was supported in part by the National Institutes of Health through the NIH Director's Early Independence Award (DP5-OD028155) and the Bill Gatton Foundation. This work was supported in part by the Kentucky Agricultural Experiment Station, Hatch project award no. 7000570, from the U.S. Department of Agriculture's (USDA) National Institute of Food and Agriculture (NIFA). The content is solely the responsibility of the authors and does not necessarily represent the official views of the sponsors.

**Competing interests:** None.

## Conclusions

Bed bug bites predominantly affect exposed extremities, although with variable dermatologic patterns that can be confused with other arthropod bites. Psychological manifestations were reported in a small subset of cases, while systemic complications were rare.

## Introduction

Bed bugs are small obligate hematophagous ectoparasites that infest human and animal dwellings [1,2]. The two primary species affecting humans are *Cimex lectularius*, and *Cimex hemipterus* [3], both with recent reports of infestations outside their historical epidemiological range [4,5]. Bed bugs practically disappeared from large regions of the more developed nations until the 1990s [6,7]. Since this time, the number of cases have risen exponentially [8], becoming a significant public health and economic burden [9]. This resurgence has been attributed to several factors including increased international travel, insecticide resistance, and changes in pest control practices [10]. Even so, there is a relative lack of scientific concern and publication of clinical and epidemiological studies of such infestations [9].

Even though bed bugs are not currently known to transmit diseases, the clinical impact of these parasites is wide. Not only can their bites produce diverse dermatological manifestations, ranging from asymptomatic reactions to severe allergic responses [9,11–14], they can also produce an important psychological impact, an aspect that has been increasingly recognized, with reports of anxiety, sleep disturbances, social stigmatization and a myriad of other manifestations [15,16]. Alongside the aforementioned health burdens, the recent discovery of bed bug-derived histamine [17,18] poises questions about the human systemic effects of such environmental contaminants, and more studies are needed to understand the possible clinical association. Having said that, current clinical presentations vary significantly among individuals, influenced by factors such as previous exposure, individual immune responses (age, genetics, amongst others), and the length and extent of infestation [9,11], although a deeper characterization is needed.

Despite these facts and their growing public health importance, a comprehensive clinical characterization of bed bug bites remains limited [10,19]. Existing literature consists primarily of isolated case reports and small case series, as well as anecdotal comments and parasitology books, with isolated clinical studies [9,20,21]. making it difficult to establish consistent clinical patterns or identify factors that might predict severity or complications.

Understanding the full spectrum of clinical manifestations is nonetheless crucial for healthcare providers, particularly given the potential for misdiagnosis and the psychological distress associated with unrecognized infestations. Moreover, as bed bug populations continue to expand and develop resistance to common insecticides [10,19], the burden on healthcare systems and patients is likely to increase, making clinical suspicion increasingly important. Therefore, this systematic review aims to synthesize available case report literature to characterize the clinical manifestations

of bed bug bites, seeking to establish a more complete understanding of their clinical spectrum and identify areas requiring further research.

## Methodology

### Search strategy

This systematic review was conducted following PRISMA guidelines. A systematic literature search was performed in multiple databases including PubMed/MEDLINE, SciELO, Bireme/LILACS, Scopus, and Google Scholar to identify all available case reports and case series of bed bug (*Cimex* spp.) infestations with clinical manifestations through November 15, 2025.

Core search strategies used included the following terms:

Complex: ("bed bug"[tiab] OR "bed bugs"[tiab] OR "Cimex lectularius"[MeSH Terms] OR "Cimex lectularius"[tiab] OR "Cimex hemipterus"[MeSH Terms] OR "Cimex hemipterus"[tiab]) AND ("case report"[Publication Type] OR "case reports"[tiab] OR "case series"[tiab] OR "case study"[tiab])

Also, a simple search strategy was employed to increase sensitivity in the search: (bed bug) AND (case report). Similar search strategies were adapted for other databases.

Although, no language restrictions were used to maximize case identification, only articles written in English, Portuguese, Spanish and French were included. Reference and citations of retrieved articles and relevant reviews were manually searched to identify additional cases not captured by electronic search.

### Inclusion and exclusion criteria

Only case reports and case series that documented human infestations caused by bed bugs were included. Eligible studies were required to provide complete clinical descriptions. Articles published in peer-reviewed journals and reports retrieved from gray literature sources (i.e., theses) were considered. Excluded papers include review articles, editorials, anecdotal evidence, blogs, and opinion pieces, as these did not present original clinical data. Experimental studies conducted in animal models and cases that lacked clear clinical descriptions, duplicate publications reporting the same patients, or reports in which the involvement of bed bugs was only speculative were not included.

### Study selection

Titles and abstracts were screened following the PRISMA guidelines. These were finally chosen using the predefined inclusion and exclusion criteria. Full-text articles were obtained for all potentially eligible studies, and the same review process was applied for final inclusion decisions. A standardized data extraction form was developed and pilot-tested on a subset of studies before full implementation.

### Data extraction

The following variables were systematically extracted from each included case. Study characteristics included the year of publication, journal, country and geographic region, study design (single case report or case series), and place of publication. Patient demographics encompassed age, sex, relevant medical history or risk factors, and the setting of exposure, whether residential, travel-related, or institutional. Clinical manifestations comprised the anatomical distribution and morphological characteristics of skin lesions, any systemic symptoms or complications, and reported psychological or behavioral manifestations. Finally, entomological data were recorded, specifically the bed bug species involved.

### Quality assessment

The quality of included case reports was assessed using a modified version of the Joanna Briggs Institute (JBI) Critical Appraisal Checklist for Case Reports and the CARE checklist [22,23]. The assessment evaluated eight domains: (1) clear

description of patient demographics and history, (2) adequate description of clinical condition, (3) clear identification of diagnostic methods, (4) appropriate diagnostic criteria, (5) adequate description of intervention or treatment, (6) clear statement of outcomes, (7) appropriate follow-up, (8) overall quality of the case details according to the CARE checklist including photographic documentation, and (9) patient perspective. The JBI domains were evaluated from 0 to 10, zero being not present and 10 being present and totally adequate. Afterwards the results of each domain were averaged and then categorized: 0–3 were categorized as low, 4–6 as medium, 7–8 as lower-upper, and 9–10 as upper. The CARE checklist was used to see the proportion of cases that reported the item in the checklist. The potential for publication bias was acknowledged. As well, the GRADE system was implemented, this system considers case reports as low-grade clinical evidence from the start.

### Statistical analysis

Data were analyzed using R statistical software (version 4.3.0). Descriptive statistics were calculated for all variables, including means and standard deviations for continuous variables and frequencies and percentages for categorical variables. Comparisons between groups (e.g., by sex) were performed. Fisher's exact tests for categorical variables and Mann-Whitney U tests for continuous variables were used, depending on data distribution and sample size. To analyze the differences between age groups, cases were stratified into four age groups: Children (0–17 years), Young Adults (18–39 years), Middle-aged Adults (40–64 years), and Older Adults (65 years and above). As multiple comparisons (k > 1) increase the experimental-wise error rate the Benjamini-Yekutieli's and Bonferroni's methods of correction were employed. The dataset that was used in this systematic review was shared as a supplemental file (S1 Dataset).

## Results

### General results

The systematic search initially identified 456 records. After removing 122 duplicates automatically and 20 duplicates manually, 314 unique records were assessed by title for screening. Of these, 186 were excluded based on title and abstract, the reasons to exclude these articles included: reviews case studies of infestations, the topic was about other parasites, arthropods, pathogens or clinical conditions, some had "bed bug" in their title but were not related, others were efficacy studies, and several were due to toxicity or deaths related to bed bug treatments. After this assessment 112 articles were fully evaluated; 48 articles were finally excluded for the following reasons: written in other languages, not enough description, lacked sufficient clinical data, were hypothetical cases, no bed bugs involved, were only focused on the entomological aspects, were just abstracts or posters, were not case reports or involved other species. Ultimately, 64 studies were included in the qualitative and quantitative synthesis as they had complete data and confirmed cases. Among the studies included, 55 were individual case reports and 9 were case series (Fig 1)

### Publication geographical origin

Regarding the place of publication, cases were reported from USA [24–41], Brazil [42–47], France [48–53], Iran [54,55] and Canada [56–59]. *C. hemipterus* was reported in Brazil [43], Iran [54,55], Colombia [14] and Turkey [60]. Other cases were reported from Finland [61], Germany [12,62,63], India [64], Israel [65], Italy [66–69], Japan [70,71], Korea [72], Mexico [73,74], Nicaragua [75], Poland [76,77], Scotland [78], Spain [79,80], Switzerland [81,82], Tunisia [83], Turkey [60], central Europe [61] and UK [84,85]. In Colombia, another reported case, presumably *C. lectularius*, acquired possibly while traveling to Nicaragua, was reported [13].

In general, the geographical origin of infestation followed the origin of the publication, having a consistent pattern: most cases appeared to be acquired locally, though the origin was unclear in some instances or possibly acquired

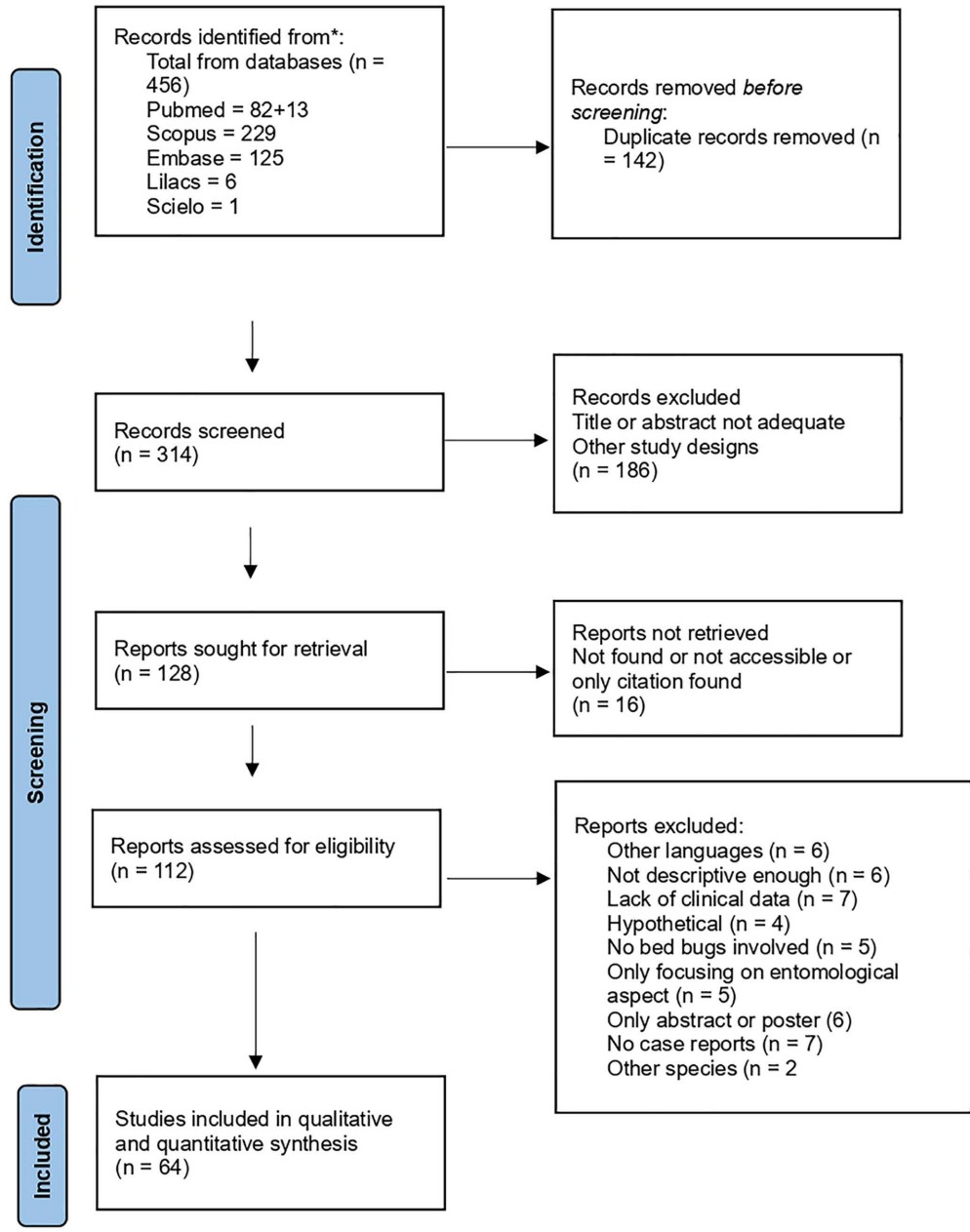

**Fig 1. PRISMA Flow Diagram.**

during travel in others [32,47,63,65,66]. However, these latter cases were relatively rare in this systematic review (7 out of 84, 8.3%).

## Sociodemographic information

The cases included in this review were reported for several reasons, for example: due to the epidemiological or geographical importance [13,14,20,24,25,28–32,34,35,40,43,48,64,66,71,78,82,86–90], association

with national or international travel [44,47,65,72,91,92], diagnosis difficulty [59,60,75,80,84], interesting or uncommon manifestations [46,57,93,94], association with a special kind of patient (i.e., neglected elderly) [45,95], and severity or complications of the case (i.e., systemic manifestations, psychological manifestations) [12,26,27,33,37,38,50,51,56,58,63,77,79,83,96–101].

From these cases, a total of 84 patients diagnosed with cimicosis caused by bed bugs (*C. lectularius* and *C. hemipterus*) were included. The sex distribution was balanced, comprising 40 females (47.6%) and 44 males (52.4%). Mean age was similar between groups: 37.5 years (SD 22.1) for females and 42.3 years (SD 21.5) for males, with no significant difference (p = 0.38). Most published cases involved *C. lectularius* (90.5%). Additionally, most cases were published between 2015–2020 (53%), with *C. hemipterus* cases appearing from 2014 onwards (Fig 2). Due to the low number of cases due to other *Cimex* species no interspecies comparisons were made.

## Location of bites

Arms (71.4%) and legs (59.5%) were commonly affected. The trunk was involved in 29 patients (34.5%; Table 1). Other bite locations included the neck, scalp [41,56], eyelids [35], and ear canal [57] (Table 1). It is important to note, that in some patients (e.g., neglected elderly), unusual locations of infestation, like clothing [38,95,102], fingernails [103,104], and prosthetic devices [41], were reported. No significant differences were observed between sexes for any anatomical location.

## Local symptoms (type of exanthem)

Papular exanthem was the most frequent presentation (76.2%) [12,37,62,63,77,79,98,100,101], with macular and bullous also reported (Table 2); one severe case of necrosis was reported [60]. The "Breakfast-Lunch-Dinner" pattern was present

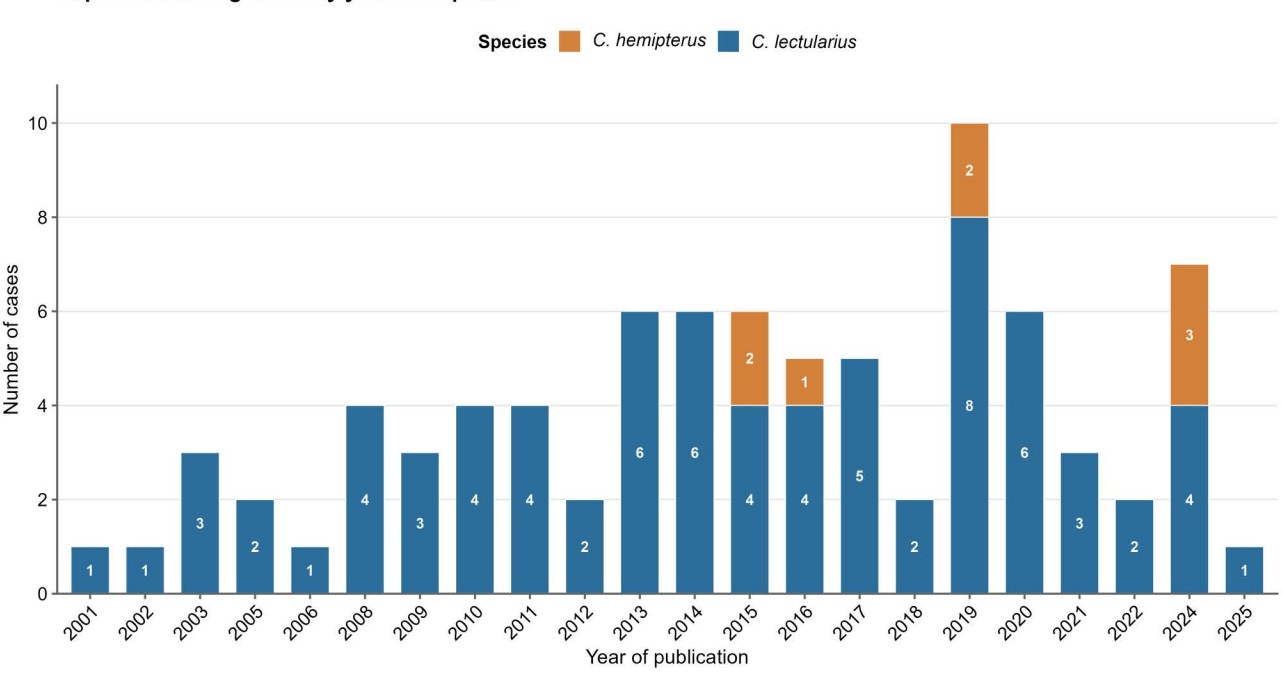

**Fig 2. Distribution of *Cimex* spp. clinical case reports by year and species.**

**Table 1. Location of bites by patient sex.**

| Location | Female (n = 40) | Male (n = 44) | Total (N = 84) | p | p_BY | p_Bonf |
|---|---|---|---|---|---|---|
| Arms | 27/40 (67.5%) | 33/44 (75.0%) | 60/84 (71.4%) | 0.73 | 0.85 | 1.00 |
| Legs | 28/40 (70.0%) | 22/44 (50.0%) | 50/84 (59.5%) | 0.08 | 0.16 | 0.57 |
| Trunk | 12/40 (30.0%) | 17/44 (38.6%) | 29/84 (34.5%) | 0.60 | 0.70 | 1.00 |
| Abdomen | 9/40 (22.5%) | 8/44 (18.2%) | 17/84 (20.2%) | 0.79 | 0.85 | 1.00 |
| Face | 11/40 (27.5%) | 5/44 (11.4%) | 16/84 (19.0%) | 0.10 | 0.23 | 0.57 |
| Back | 8/40 (20.0%) | 8/44 (18.2%) | 16/84 (19.0%) | 1.00 | 1.00 | 1.00 |
| Neck | 8/40 (20.0%) | 5/44 (11.4%) | 13/84 (15.5%) | 0.40 | 0.58 | 1.00 |
| Buttocks | 6/40 (15.0%) | 5/44 (11.4%) | 11/84 (13.1%) | 0.83 | 0.93 | 1.00 |
| Eyelids | 2/40 (5.0%) | 3/44 (6.8%) | 5/84 (6.0%) | 1.00 | 1.00 | 1.00 |
| Inguinal | 2/40 (5.0%) | 3/44 (6.8%) | 5/84 (6.0%) | 1.00 | 1.00 | 1.00 |
| Hands | 0/40 (0.0%) | 4/44 (9.1%) | 4/84 (4.8%) | 0.16 | 0.31 | 0.79 |
| Scalp | 0/40 (0.0%) | 2/44 (4.5%) | 2/84 (2.4%) | 0.53 | 0.70 | 1.00 |
| Ear canal | 0/40 (0.0%) | 1/44 (2.3%) | 1/84 (1.2%) | 1.00 | 1.00 | 1.00 |

*p = p-value comparing incidence between sexes using Fisher's exact test; p_BY = p-value corrected by the Benjamini-Yekutieli method; p_Bonf = p-value corrected by the Bonferroni method.

in less than half the patients (females 47.5% vs males 43.2%, p value = 0.787). There were no differences between sexes (Table 2).

## Psychological and psychiatric symptoms

Psychological symptoms included anxiety, stress, and hopelessness (9.5%) [14,32,36,55,58] and sleep deprivation, the latter reported exclusively in females (15.0%), albeit not reaching statistical significance after correction [14,32,36]. (Table 3). Bed bugs have been implied in cases of delusional parasitosis (also known as Ekbom syndrome) [26,105]. Other results can be seen in Table 3.

**Table 2. Exanthem types reported in the clinical case reports.**

| Type of exanthem | Female (n = 40) | Male (n = 44) | Total (N = 84) | p | p_BY | p_Bonf |
|---|---|---|---|---|---|---|
| Papular | 30/40 (75.0%) | 34/44 (77.3%) | 64/84 (76.2%) | 1.00 | 1.00 | 1.00 |
| Macular | 24/40 (60.0%) | 21/44 (47.7%) | 45/84 (53.6%) | 0.31 | 0.58 | 1.00 |
| Linear | 19 (47.5%) | 19 (43.2%) | 38 (45.2%) | 0.85 | 1.00 | 1.00 |
| Bullous | 4/40 (10.0%) | 8/44 (18.2%) | 12/84 (14.3%) | 0.47 | 0.70 | 1.00 |
| Blister | 4/40 (10.0%) | 5/44 (11.4%) | 9/84 (10.7%) | 1.00 | 1.00 | 1.00 |
| Hemorrhagic | 4/40 (10.0%) | 5/44 (11.4%) | 9/84 (10.7%) | 1.00 | 1.00 | 1.00 |
| Targetoid | 1/40 (2.5%) | 0/44 (0.0%) | 1/84 (1.2%) | 0.95 | 1.00 | 1.00 |
| Petechial plaques | 1/40 (2.5%) | 0/44 (0.0%) | 1/84 (1.2%) | 0.95 | 1.00 | 1.00 |

*p = p-value comparing incidence between sexes using Fisher's exact test; p_BY = p-value corrected by the Benjamini-Yekutieli method; p_Bonf = p-value corrected by the Bonferroni method.

**Table 3. Psychological and psychiatric symptoms.**

| Symptom | Female (n = 40) | Male (n = 44) | Total (N = 84) | p | p_BY | p_Bonf |
|---|---|---|---|---|---|---|
| Anxiety | 7/40 (17.5%) | 1/44 (2.3%) | 8/84 (9.5%) | 0.04 | 0.13 | 0.57 |
| Stress | 5/40 (12.5%) | 3/44 (6.8%) | 8/84 (9.5%) | 0.58 | 0.70 | 1.00 |
| Hopelessness | 6/40 (15.0%) | 2/44 (4.5%) | 8/84 (9.5%) | 0.20 | 0.39 | 0.79 |
| Sleep deprivation | 6/40 (15.0%) | 0/44 (0.0%) | 6/84 (7.1%) | 0.02 | 0.12 | 0.41 |
| Anger | 2/40 (5.0%) | 4/44 (9.1%) | 6/84 (7.1%) | 0.78 | 0.85 | 1.00 |
| Depression | 4/40 (10.0%) | 0/44 (0.0%) | 4/84 (4.8%) | 0.10 | 0.22 | 0.57 |
| Incredulity | 1/40 (2.5%) | 1/44 (2.3%) | 2/84 (2.4%) | 1.00 | 1.00 | 1.00 |
| Alcohol abuse | 2/40 (5.0%) | 0/44 (0.0%) | 2/84 (2.4%) | 0.42 | 0.58 | 1.00 |
| Suicide ideation | 2/40 (5.0%) | 0/44 (0.0%) | 2/84 (2.4%) | 0.42 | 0.58 | 1.00 |
| Suicide attempt | 2/40 (5.0%) | 0/44 (0.0%) | 2/84 (2.4%) | 0.42 | 0.58 | 1.00 |
| Mood lability | 2/40 (5.0%) | 0/44 (0.0%) | 2/84 (2.4%) | 0.42 | 0.58 | 1.00 |
| Embarrassment | 0/40 (0.0%) | 1/44 (2.3%) | 1/84 (1.2%) | 1.00 | 1.00 | 1.00 |
| Financial distress | 1/40 (2.5%) | 0/44 (0.0%) | 1/84 (1.2%) | 0.95 | 1.00 | 1.00 |

*p = p-value comparing incidence between sexes using Fisher's exact test; p_BY = p-value corrected by the Benjamini-Yekutieli method; p_Bonf = p-value corrected by the Bonferroni method.

## Systemic manifestations

Interestingly, several systemic symptoms were reported in association with bed bug bites, some of which occurring in cases with anemia [27,38,41,50,56,95,102,106]. These and the severe allergic reactions and vasculitis reported in a small number of patients can be seen in Table 4 [41,56,81,95,96]. No sex-based differences were observed.

## Analysis by age

Several age-related differences were identified in both cutaneous and systemic findings. Facial involvement was markedly more frequent in children (60.0%) compared to young adults (18.2%), middle-aged adults (7.4%), and older adults (16.7%) (p_Bonf = 0.345). Eyelid involvement was reported exclusively in children and absent in all other age groups (p_Bonf = 0.007). Conversely, buttock involvement was predominantly observed in older adults (50.0%), with considerably lower rates across children (10.0%), young adults (12.1%), and middle-aged adults (0.0%) (p_Bonf = 0.007). Trunk and back involvement followed intermediate patterns, with children generally more affected than young adults, though older adults showed increased involvement in certain areas (trunk: p = 0.024; back: p_ = 0.014, neither significant after correction). Anemia was reported exclusively in middle-aged (14.8%) and older adults (25.0%), being entirely absent in children and young adults (p = 0.025, not significant after correction). No other variables differed significantly across age groups. Finally, many of the psychological symptoms were reported more commonly in young adults (S1 File).

## Quality assessment

Reports quality was variable but low. Overall quality assessment was modest: 30.6% were rated as poor, 28.2% medium, 35.3% lower-upper, and 5.9% upper quality. Most studies provided limited demographic information, and clinical descriptions of lesions and general patient status were often incomplete or inconsistent. Standardized measures and long-term follow-up data were scarce. The systematic evaluation of psychological or behavioral impacts was uncommon (7.4% of reports). In addition, information regarding treatment of both medical symptoms and infestation management was frequently sparse or poorly described. Using the GRADE system 94.1% of the cases were Very Low Level of Evidence and 5.9% were classified as Low Level of Evidence.

**Table 4. Systemic symptomatology.**

| Systemic sign/symptom | Female (n = 40) | Male (n = 44) | Total (N = 84) | p | p_BY | p_Bonf |
|---|---|---|---|---|---|---|
| Anemia | 2/40 (5.0%) | 5/44 (11.4%) | 7/84 (8.3%) | 0.53 | 0.70 | 1.00 |
| Dizziness | 2/40 (5.0%) | 3/44 (6.8%) | 5/84 (6.0%) | 1.00 | 1.00 | 1.00 |
| Fatigue | 1/40 (2.5%) | 3/44 (6.8%) | 4/84 (4.8%) | 0.70 | 0.85 | 1.00 |
| Lethargy | 1/40 (2.5%) | 3/44 (6.8%) | 4/84 (4.8%) | 0.70 | 0.85 | 1.00 |
| Tachycardia | 2/40 (5.0%) | 1/44 (2.3%) | 3/84 (3.6%) | 0.92 | 1.00 | 1.00 |
| Cough | 1/40 (2.5%) | 1/44 (2.3%) | 2/84 (2.4%) | 1.00 | 1.00 | 1.00 |
| Fever | 1/40 (2.5%) | 1/44 (2.3%) | 2/84 (2.4%) | 1.00 | 1.00 | 1.00 |
| Hyperpigmentation | 2/40 (5.0%) | 0/44 (0.0%) | 2/84 (2.4%) | 0.42 | 0.58 | 1.00 |
| Flares | 0/40 (0.0%) | 1/44 (2.3%) | 1/84 (1.2%) | 1.00 | 1.00 | 1.00 |
| Malaise | 0/40 (0.0%) | 1/44 (2.3%) | 1/84 (1.2%) | 1.00 | 1.00 | 1.00 |
| Paleness | 0/40 (0.0%) | 1/44 (2.3%) | 1/84 (1.2%) | 1.00 | 1.00 | 1.00 |
| Dyspnea | 0/40 (0.0%) | 1/44 (2.3%) | 1/84 (1.2%) | 1.00 | 1.00 | 1.00 |

*p = p-value comparing incidence between sexes using Fisher's exact test; p_BY = p-value corrected by the Benjamini-Yekutieli method; p_Bonf = p-value corrected by the Bonferroni method.

## Discussion

Blood feeding has evolved in multiple families of arthropods with several modes of blood acquisition [107]. While bites can have different clinical characteristics, clinical diagnosis is often difficult given the number of species capable of blood feeding [107]. To further complicate things, the clinical characteristics can be confused with a myriad of other dermatological conditions, systemic diseases, and the defensive bites and stings of other arthropods [108]. Clinically, attempts have been made to try to differentiate arthropod bites, and even when differentiating between telmophagy (creating blood pools) and solenophagy (siphoning blood) can be possible, in real clinical practice, without a thorough medical history and known epidemiology, it is often not possible to do so for most arthropods bites, including those due to bed bugs. As such, it must be highlighted that several other arthropods can produce similar clinical patterns, including the severe manifestations [109,110] found in these cases. In most arthropod bites dermatological reactions predominate; thus, there is a prominent clinical similarity between bed bugs and other hematophagous insects.

Even so, the characterization of such bites and their associated complications is essential. In this systematic review, 64 reports related to bed bugs were included. This allowed the synthesis of 85 patients with bed bugs. Although limited and biased, this review provides a characterization of clinical, anatomical, and psychological manifestations of bed bug bite cases. While prior literature describes the clinical spectrum of bed bug bites the present work advances this knowledge in several ways: it provides quantitative estimation of symptom and presentation frequencies; uses nonparametric statistical comparisons to formally test differences across sex groups; and explores age as a potential modifier of clinical presentation.

Regarding the place of publication, cases were reported from all over the world with cases from Asia, Europe, North America, Central America and South America; no cases were captured from Africa. Even so, publication and language bias over represented the USA [24–41], Brazil [42–47] and France [48–53], which limited not only the type of patients, but also the reasons of publication, and the bed bug species involved. On the other hand, this review was able to capture just a handful of tropical bed bug associated cases. Such reports were obtained from Brazil [43], Iran, Colombia [14] and Turkey. Having said that, it is important to highlight that the location of infestation followed the origin of the publication. As well, it is also significant to note that case reports often focus on epidemiological or geographical novelties; as such, these would be expected to be higher in developed countries and rarer in developing countries, as bed bug infestations are negatively correlated with income [111]

When evaluating who bed bugs affect, a balanced sex distribution with a similar age range was found, supporting the notion that bed bugs affect people irrespective of age or gender [3,112–117]. In this review, the assessment of socioeconomical status was difficult; however, it is important to note that even if everyone can get bed bugs, the ones that are affected the most and for the longest time are individuals of lower socioeconomical status, this lies on the fact that the lower socioeconomical strata do not have the economic resources necessary to treat the infestation on their own and can have more risk of acquiring them [118]. Even so, some differences between age groups were observed in several skin and systemic findings. Facial involvement was reported more in children than in other age groups; this difference could reflect publication bias or the accessibility of the bed bugs to the patient's face (infants can be bundled up or covered so bed bugs don't have access to the limbs). Interestingly this involvement followed a similar pattern found in patients with Chagas disease. In some cases triatomine bugs (also known as kissing bugs, the Chagas disease vector) [119], feed near the eyes and cause what is known as the Romaña sign, this is also commonly present in children [119]. Buttocks involvement was highest in older adults and very low in other age groups, possibly reflecting different sleeping practices as sleeping in certain types of underwear or naked. Finally, systemic involvement, such as anemia, was reported only in middle-aged and older adults and absent in younger groups, which could reflect more body fragility as patients grew older or reflecting cases of neglect or lack of self-care [50,102].

Based on the available case reports, bed bugs bit several different parts of the body. Anatomically, extremities appeared to be most affected, followed by the trunk, consistent with previous articles and published works and bed bug biology [9,19,120], and is also relatively similar to the ones of mosquitos and fleas, although mosquito bites tend to be more scattered and fleas more in the lower extremities and not upper ones (although *Aedes* mosquitos also tend to bite along the ankle) [107]. Additional bite locations included the eyelids, scalp, neck, and ear canal, which may reflect the accessibility of these areas during nocturnal feeding [9,19,120]. The so-called distinctive "Breakfast-Lunch-Dinner" linear pattern was observed in less than 50% of patients, which has been attributed to sequential feeding behavior [9,19,43,120]. This pattern is not unique to bed bugs, as it has been reported in some patients with dermatological reactions and allergies, other arthropods, including kissing bugs [121], fleas [122], among others. This calls into question how clinically useful or pathognomonic of the genera this pattern really is.

Regarding lesion morphology, the most common lesions were simple and not complicated. Papular lesions, that are commonly reported after insect bites, appeared to be most common, followed by macular, which can be the second stage of the former and interestingly, more severe presentations (as bullous lesions) were reported in almost 2 out of 10 patients, although this could reflect the publication bias of case reports. Other patterns were included (i.e., petechial and targetoid lesions) but notably, crusts and excoriations were infrequently reported. This lack of report could be due to memory or importance bias.

Systemic symptoms were also reported; when present they could be severe or produce complications in fragile patients. One of these documented symptoms was anemia. This sign, and other systemic signs and symptoms, reported in cases of *C. lectularius*, were associated with blood loss. Severe allergic or immunological reactions (such as vasculitis) were reported in a small number of patients, suggesting that, while most bed bug bites remain primarily dermatological, systemic complications do occur. These signs and symptoms occurred in patients who were more economically, psychologically or socially fragile [26,95], especially older patients and in patients who had concomitant psychological conditions. Also, these patients can live in neglected conditions

These findings show an interesting contrast with the broader literature, which highlights the substantial psychological burden of bed bug infestations. Our findings have important clinical implications. First is that possible clinical patterns were observed across cases, suggesting that a subgroup of patients tends to develop more severe psychiatric and psychological manifestations; however, larger prospective cohorts are needed to corroborate whether these observed patterns are true [15,16,36]. The lack of certainty on this aspect can be seen in the under reporting of psychological and psychiatric symptoms in the included cases; on the other hand, a scoping review of 51 papers [15] (original, literature

review, technical guidelines, among others) found that nearly three-quarters reported general psychological distress, one-third reported diagnosable psychiatric disorders, and a smaller fraction reported severe outcomes such as psychosis or suicidality. Commonly reported symptoms include insomnia, anxiety, panic, hypervigilance, and PTSD-like features, often exacerbated by the chronic difficulty of eradication [16]. Case reports described worsening of pre-existing psychiatric conditions, and, in extreme cases, suicide linked to repeated infestations, which was also included in this review [58]. Indirect consequences such as stigma, social isolation, and financial burden further amplify psychological morbidity, particularly among vulnerable populations, including the elderly, people experiencing homelessness, and those with pre-existing mental illness. Taken together, our results and the published evidence underscore that the psychological sequelae of bed bug infestations can be profound, and underreported [15]. These potential psychological and social impacts may require supportive interventions beyond dermatological treatment [15]. Ultimately, large retrospective and prospective clinical studies are essential to provide more robust data on the true frequency and severity of different manifestations, while controlled studies comparing presentations across populations could help identify risk factors for severe reactions and provide valuable insights for comprehensive patient care.

This systematic review has several important limitations that should be considered when interpreting the findings. The reliance on published case reports introduces inherent publication bias, as unusual, severe, or complex presentations are more likely to be reported than typical cases, potentially overestimating the frequency of complications and atypical manifestations. As such, these findings are best interpreted as hypothesis-generating rather than definitive. The retrospective nature of case reports limits the standardization of clinical assessments, with significant variability in the depth and quality of clinical documentation across studies. Also, some cases are based primarily on clinical presentation and circumstantial evidence of bed bug presence so other causes cannot be ruled out, potentially introducing noise. The geographical diversity and a relatively small subset of patients with different environments can also contribute noise to the results. The cross-sectional nature of most case reports provides limited insight into the natural history, temporal progression, seasonality or long-term outcomes of bed bug-related manifestations. The assessment of psychological symptoms was particularly limited by inconsistent reporting and a small subset of cases reported. It is important to remember, that case reports frequently emphasize the most striking clinical features, are often based on evaluations conducted several months after disease onset, or selectively describe more severe manifestations. The lesions and other clinical manifestations summarized in this review may be influenced both by publication bias (favoring unusual or severe cases), information (patient and physician recall), severity, and by temporal bias related to the timing of clinical evaluation. Taken together, these biases underscore the need for more rigorous and standardized reporting in future studies

In conclusion, the bed bug cases included here suggest an overall relatively benign course, especially when considering the fact that case reports tend to report on unusual presentations. That said, there remains the possibility of complications, including unreported psychological symptomatology. Furthermore, given the diversity of dermatological symptoms, bite morphology is an unreliable tool in diagnosing bed bug infestations, with presentations often indistinguishable from other dermatological conditions, including bites from other arthropods. However, when maculopapular pruriginous lesions on extremities are found in patients lacking other plausible explanations, inspection of the home for bed bugs may be worth considering. This systematic review offers a preliminary foundation for evidence-based recognition, while underscoring the need for larger prospective studies to further characterize the clinical spectrum and improve care for affected patients.

## Keypoints

- Question: What are the demographic, clinical, and epidemiologic characteristics of bed bug infestations reported in the literature?

- Findings: In this systematic review of 84 patients, most cases were published between 2015 and 2020, predominantly involved *C. lectularius* (90.5%), and were acquired locally. Dermatologic manifestations were most commonly papular

(76.2%), followed by macular (53.6%), which sometimes followed a linear pattern(45.2%), with extremities and trunk being the most affected sites. Systemic symptoms were uncommon, but psychological impact was observed in some patients.

- Meaning: Bed bug infestations present with variable dermatologic patterns and with adiverse and heterogenous subet of systemic psychological sequelae. Recognition of atypical lesion locations and consideration of patient vulnerability are important but neither sufficient nor necessary signs of bed bug infestation or bites diagnosis.

## Supporting information

**S1 File. Supplement 2 for bed bug article.**
(DOCX)

**S1 Dataset. Bedbug minimal dataset.**
(XLSX)

**S1 Checklist. ISSM MOOSE Checklist.**
(PDF)

**S2 Checklist. PRISMA 2020 checklist BB.**
(DOCX)

## Author contributions

**Conceptualization:** Julian Felipe Porras-Villamil, Zachary C. DeVries.

**Data curation:** Julian Felipe Porras-Villamil, Zachary C. DeVries.

**Formal analysis:** Julian Felipe Porras-Villamil, Zachary C. DeVries.

**Funding acquisition:** Zachary C. DeVries.

**Investigation:** Julian Felipe Porras-Villamil, Zachary C. DeVries.

**Methodology:** Julian Felipe Porras-Villamil, Zachary C. DeVries.

**Project administration:** Julian Felipe Porras-Villamil, Zachary C. DeVries.

**Resources:** Julian Felipe Porras-Villamil, Zachary C. DeVries.

**Software:** Julian Felipe Porras-Villamil.

**Supervision:** Julian Felipe Porras-Villamil, Zachary C. DeVries.

**Validation:** Julian Felipe Porras-Villamil.

**Visualization:** Julian Felipe Porras-Villamil.

**Writing – original draft:** Julian Felipe Porras-Villamil, Zachary C. DeVries.

**Writing – review & editing:** Julian Felipe Porras-Villamil, Zachary C. DeVries.

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
