## [Decision Letter · Decision Letter 0]

17 Feb 2026

PLOS One

Dear Dr. Porras Villamil,

Thank you for submitting your manuscript to PLOS ONE.  Your work has now been reviewed by two-peer experts and myself. After careful consideration, we feel that it has merit but does not fully meet PLOS ONE’s publication criteria as it currently stands. Therefore, we invite you to submit a revised version of the manuscript that addresses the points raised during the review process.

We look forward to receiving your revised manuscript.

Kind regards,

Jose Pietri

Academic Editor

PLOS One

Additional Editor Comments:

Reviewers' comments:

Reviewer's Responses to Questions

**Comments to the Author**

1. Is the manuscript technically sound, and do the data support the conclusions?

Reviewer #1: Partly

Reviewer #2: Yes

2. Has the statistical analysis been performed appropriately and rigorously?

Reviewer #1: Yes

Reviewer #2: Yes

3. Have the authors made all data underlying the findings in their manuscript fully available?

Reviewer #1: Yes

Reviewer #2: Yes

4. Is the manuscript presented in an intelligible fashion and written in standard English?

Reviewer #1: No

Reviewer #2: Yes

Reviewer #1:

This manuscript presents a review of bedbug case reports. The goal was to describe the clinical spectrum of bed-bug related health impacts. From the identified and included case reports, the authors extracted data and attempted a meta-analysis of sorts.

This clearly was a lot of work; but trying secondary analysis from the extremely-heterogeneous case report format is difficult. I think their findings are probably worth publishing, but they are not really as "evidence-based" as the authors seem to think.

There is little new in this paper compared with clinical descriptions already out there (e.g. such as a standard text, reference #2 in this manuscript). While one would like to think the present work is a systematic confirmation of what the textbooks and experts state is so, it is a systematic review of *uncontrolled observations* so basically doesn't really improve the confidence with which one can diagnose the bites.

Should it be published, I think the most important thing for the revision is to make it *more concise*. Although it is within the general word limits, it will be MUCH more effective if it is concise. There is plenty of redundancy in the current version so this should be easily attainable.

COMMENTS/SUGGESTIONS (by line number)

43 This line is redundant (“Psychological manifestations... Psychological symptoms...”). Please simplify.

44 “...more women than men” isn’t supported by the data in Table 3. Consider eliminating this point from the ABSTRACT

161 It is not clear which, if any, of the data presented in the tables is Gaussian (bite location, rash type, psychological or systemic symptoms); consider sticking with the nonparametric statistical tools.

165 Using p<0.05 where there are multiple comparisons (k>1) increases the experimental-wise error rate. Consider deleting this sentence. In its place, a brief mention of the use of Benjamini-Yekutieli’s and Bonferroni’s methods, which are later presented and abbreviated without explanation, would be good. The authors are to be commended for taking into account experiment-wise error rate using these tools. Such consideration is neglected in far too many research reports.

197 Consider using a different word for “location” in this context (or precede it by “geographical”); “location” is used later to indicate anatomy, rather than geography. Presumably the bite site isn’t specific to publication location.

218-257 The narrative here could be greatly shortened. There is no need to repeat what is in Tables 1-4 in the text; highlight a couple interesting points and refer to the table. This will greatly improve readability of the paper.

220 The long list of citations inserted after each point feels redundant and may not be necessary.

331 It might be worth mentioning that in the higher income countries from which most of the case reports originated, bedbugs are a bit of a novelty; in many lower income regions they are so prevalent as to be a fact of life and don’t warrant reporting in the minds of the clinicians.

342 Consider expanding on what is intended by the connection to Triatoma. Also, at 358 “kissing bugs” is used. Non-entomologists may not recognize these are being used synonymously, so consider using one term or the other in both places.

376 “Lack of hygiene” may not be entirely correct; often it is the presence of clutter, at least in higher income nations. Cf. https://www.baycountymi.gov/Docs/Bed%20Bugs%20Fact%20Sheet.pdf

420 This is an important limitation: case reports are NOT typical cases, they are unusual and therefore hypothesis generating, not data producing. A review of case reports in an effort to learn what is normal is fraught with peril. A study of “zebras” may not offer accurate insight into “horses”. This study is a valiant and worthwhile effort, but the nature of the tasks suggests that the conclusion at 450 is perhaps overly bold.

465ff The REFERENCE section requires significant copy-editing for consistency. Variable strategies of capitalization, abbreviation, bibliographic information included, and format of the references are used. This section appears to have been imported directly into the manuscript from bibliography software without review and revision to ensure consistency or compliance with journal requirements.

861 There is some redundancy between the main text and what is included in the boxes in the figure. This reviewer likes seeing it included in the figure, so would probably trim it out of the narrative.

866 Consider increasing the font size in Figure 2: it is barely readable in the review copy; once reduced for publication the labels and legend will be nearly invisible.

878 In the first column, it is not clear what the “(%)” is intended to indicate; it may be an artifact from an earlier draft – subsequent columns include both “n” and “%” appropriately

OVERALL

The manuscript is very lengthy for the amount of data it conveys. The narrative includes a significant amount of redundancy. The paper will be much more effective if it is reduced to approximately half of its current word count. Some of the suggestions will assist toward this goal.

In addition, the main narrative would benefit from minor copyediting for paragraph and section breaks, spelling, punctuation, and consistency. As one example of the latter, compare the parenthetical phrases at 230 vs. 231.

Reviewer #2:

1- The paper entitled "Clinical Manifestations of Bed Bug Bites: A Systematic Review of Case Reports" by Porras-Villamil et al. interested me and confirms many observations I've made since I started studying this insect. Regarding the format, just one line 271, should have a line break for "Unusual locations".

2 - I don't see the point in taking the species Cimex pipistrelli into account since there is only one reported case.Or, if it is considered, other species like *Cimex hirundinis* should be added, of which at least one case exists :

- Principato M, Iolanda M, Principato S, Lanza F & Stingeni L, 2018 – Occupational human infestation due to “martin bug” (Oeciacus hirundinis, Hemiptera: Cimicidae). International Journal of Dermatology, 58: 115-116.

3 - About 'facial involvement", line 340-341, I have often observed this phenomenon, and I believe it is due to the fact that babies sleep in sleeping bags and toddlers are very often well covered when they sleep. Only the head is accessible for bed bugs.

4- Finally, it would have been interesting to consider, if available, seasonal data, especially for temperate climates, to see the distribution of bites throughout the year.

.

Reviewer #1: No

Reviewer #2: **Yes:** BERENGER Jean-MichelBERENGER Jean-MichelBERENGER Jean-MichelBERENGER Jean-Michel

---

## [Author Response · Author response to Decision Letter 1]

14 Mar 2026

Response to Reviewers

Manuscript ID: PONE-D-25-69185

Title: Clinical Manifestations of Bed Bug Bites: A Systematic Review of Case Reports

We sincerely thank the Academic Editor and both reviewers for their thoughtful and constructive feedback. We have addressed all comments and believe the revised manuscript is substantially improved. Below we detail our responses point by point.

Academic Editor

Editor's comment:

I advise you to pay particular attention to addressing the main concerns of reviewer 1 regarding redundancy within the text. I also suggest that the advancements that this paper provides over similar past works that have compiled information on clinical reports of bed bugs should be more strongly spelled out and highlighted.

Response: We have substantially reduced redundancy throughout the manuscript, particularly in the Results section (lines 218–257), the Abstract (lines 42–45), and the Figure 1 narrative. We have also added a more explicit discussion highlighting how this systematic review differs from and extends prior compilations (Lines 322–325), including narrative textbook descriptions (reference #2), by providing a structured, reproducible synthesis across a large number of case reports with formal statistical analysis.

Reviewer #1

General comments:

This manuscript presents a review of bedbug case reports. The goal was to describe the clinical spectrum of bed-bug related health impacts. From the identified and included case reports, the authors extracted data and attempted a meta-analysis of sorts.

This clearly was a lot of work; but trying secondary analysis from the extremely-heterogeneous case report format is difficult. I think their findings are probably worth publishing, but they are not really as "evidence-based" as the authors seem to think.

There is little new in this paper compared with clinical descriptions already out there (e.g. such as a standard text, reference #2 in this manuscript). While one would like to think the present work is a systematic confirmation of what the textbooks and experts state is so, it is a systematic review of uncontrolled observations so basically doesn't really improve the confidence with which one can diagnose the bites.

Should it be published, I think the most important thing for the revision is to make it more concise. Although it is within the general word limits, it will be MUCH more effective if it is concise. There is plenty of redundancy in the current version so this should be easily attainable.

Response: We thank Reviewer 1 for the detailed line-by-line comments. This feedback was particularly helpful in tightening the manuscript. We have reduced the pages count by condensing the Results and modifying briefly the introduction and discussion. On the other hand, part of the effort that was carried out was to show that in uncontrolled observations it is difficult to diagnose bed bugs solely on symptomatology and bite morphology. As this is the case, and as was stated in your comment due to the heterogeneous case report quality and data reporting, in uncontrolled settings, clinical awareness and confirmation are essential.

Comment 1 (Line 43 – Abstract redundancy):

43 This line is redundant ("Psychological manifestations... Psychological symptoms..."). Please simplify.

Response: Revised. The redundant phrasing regarding psychological manifestations has been simplified. (Lines 42–45)

Comment 2 (Line 44 – Sex comparison in Abstract):

44 "...more women than men" isn't supported by the data in Table 3. Consider eliminating this point from the ABSTRACT.

Response: This point has been removed from the Abstract.

Comment 3 (Line 161 – Gaussian assumption):

161 It is not clear which, if any, of the data presented in the tables is Gaussian (bite location, rash type, psychological or systemic symptoms); consider sticking with the nonparametric statistical tools.

Response: We agree. We have revised this section to rely on nonparametric tools throughout, given the nature of the data. (Lines 155–157)

Comment 4 (Line 165 – Multiple comparisons):

165 Using p<0.05 where there are multiple comparisons (k>1) increases the experimental-wise error rate. Consider deleting this sentence. In its place, a brief mention of the use of Benjamini-Yekutieli's and Bonferroni's methods, which are later presented and abbreviated without explanation, would be good. The authors are to be commended for taking into account experiment-wise error rate using these tools. Such consideration is neglected in far too many research reports.

Response: The sentence referencing p<0.05 as a blanket threshold has been removed. In its place, we now briefly introduce the Benjamini-Yekutieli and Bonferroni correction methods at first mention, so readers understand the rationale before encountering the abbreviations. (Lines 161–162)

Comment 5 (Line 197 – "location" ambiguity):

197 Consider using a different word for "location" in this context (or precede it by "geographical"); "location" is used later to indicate anatomy, rather than geography. Presumably the bite site isn't specific to publication location.

Response: We have replaced "location" with "geographical origin" when referring to publication country, to avoid confusion with anatomical bite site used later in the text. (Line 182)

Comment 6 (Lines 218–257 – Narrative redundancy):

218-257 The narrative here could be greatly shortened. There is no need to repeat what is in Tables 1-4 in the text; highlight a couple interesting points and refer to the table. This will greatly improve readability of the paper.

Response: This section has been substantially condensed. Rather than repeating table contents in prose, we now highlight only the most notable findings and direct readers to the relevant tables. (Lines 217–296)

Comment 7 (Line 220 – Citation lists):

220 The long list of citations inserted after each point feels redundant and may not be necessary.

Response: Long citation strings have been eliminated.

Comment 8 (Line 331 – Under-reporting in endemic regions):

331 It might be worth mentioning that in the higher income countries from which most of the case reports originated, bedbugs are a bit of a novelty; in many lower income regions they are so prevalent as to be a fact of life and don't warrant reporting in the minds of the clinicians.

Response: We have added a sentence acknowledging that in lower-income regions where bed bugs are highly prevalent, clinicians may not consider individual cases worth reporting, which likely contributes to the geographic bias in our dataset. (Lines 338–340)

Comment 9 (Lines 342/358 – Triatoma / kissing bugs):

342 Consider expanding on what is intended by the connection to Triatoma. Also, at 358 "kissing bugs" is used. Non-entomologists may not recognize these are being used synonymously, so consider using one term or the other in both places.

Response: We have expanded the Triatoma bite presentation similarity in the discussion. As well we have standardized terminology; “kissing bugs" is now used consistently in both places with a parenthetical clarification on first use. We have as well briefly expanded the connection to Triatoma

Comment 10 (Line 376 – "Lack of hygiene"):

376 "Lack of hygiene" may not be entirely correct; often it is the presence of clutter, at least in higher income nations. Cf. https://www.baycountymi.gov/Docs/Bed%20Bugs%20Fact%20Sheet.pdf

Response: We have revised this to reflect current evidence more accurately, noting that clutter and high-turnover environments are more strongly associated with infestation than hygiene per se. (Lines 404–407). Even so, the presence of bed bugs on the patients themselves highlights the neglect and lack of hygiene in those patients.

Comment 11 (Line 420 – Case report limitations):

420 This is an important limitation: case reports are NOT typical cases, they are unusual and therefore hypothesis generating, not data producing. A review of case reports in an effort to learn what is normal is fraught with peril. A study of "zebras" may not offer accurate insight into "horses". This study is a valiant and worthwhile effort, but the nature of the tasks suggests that the conclusion at 450 is perhaps overly bold.

Response: We have strengthened this limitation paragraph and tempered the concluding statement at lines 474–491 accordingly, explicitly noting that case reports represent atypical presentations and that findings should be considered hypothesis-generating rather than definitive.

Comment 12 (Lines 465ff – References):

465ff The REFERENCE section requires significant copy-editing for consistency. Variable strategies of capitalization, abbreviation, bibliographic information included, and format of the references are used. This section appears to have been imported directly into the manuscript from bibliography software without review and revision to ensure consistency or compliance with journal requirements.

Response: The reference list has been thoroughly reviewed and reformatted for consistency in capitalization, abbreviation, and PLOS ONE style.

Comment 13 (Line 861 – Figure/text redundancy):

861 There is some redundancy between the main text and what is included in the boxes in the figure. This reviewer likes seeing it included in the figure, so would probably trim it out of the narrative.

Response: Overlapping narrative has been trimmed from the main text in favor of keeping the information within the figure, as the reviewer suggested. (Lines 172–176 and Figure 1)

Comment 14 (Line 866 – Figure 2 font size):

866 Consider increasing the font size in Figure 2: it is barely readable in the review copy; once reduced for publication the labels and legend will be nearly invisible.

Response: Font size has been increased throughout Figure 2 to improve legibility, and the format has been improved.

Comment 15 (Line 878 – Table column header artifact):

878 In the first column, it is not clear what the "(%)" is intended to indicate; it may be an artifact from an earlier draft – subsequent columns include both "n" and "%" appropriately.

Response: The erroneous "(%)" artifact has been removed from the first column headers.

Overall comment:

The manuscript is very lengthy for the amount of data it conveys. The narrative includes a significant amount of redundancy. The paper will be much more effective if it is reduced to approximately half of its current word count. Some of the suggestions will assist toward this goal.

In addition, the main narrative would benefit from minor copyediting for paragraph and section breaks, spelling, punctuation, and consistency. As one example of the latter, compare the parenthetical phrases at 230 vs. 231.

Response: We have substantially reduced the overall word count by condensing the Results narrative, eliminating redundant citation strings, trimming the figure narrative overlap, tweaking Introduction and Discussion, and simplifying the Abstract. The manuscript has also been copyedited for paragraph breaks, punctuation, spelling, and internal consistency throughout.

Reviewer #2

Comment 1 (Line 271 – Line break):

1- The paper entitled "Clinical Manifestations of Bed Bug Bites: A Systematic Review of Case Reports" by Porras-Villamil et al. interested me and confirms many observations I've made since I started studying this insect. Regarding the format, just one line 271, should have a line break for "Unusual locations".

Response: The line break for "Unusual locations" has been added. It was relocated to lines 197–200.

Comment 2 (Cimex pipistrelli / Cimex hirundinis):

2 - I don't see the point in taking the species Cimex pipistrelli into account since there is only one reported case. Or, if it is considered, other species like Cimex hirundinis should be added, of which at least one case exists:

• Principato M, Iolanda M, Principato S, Lanza F & Stingeni L, 2018 – Occupational human infestation due to "martin bug" (Oeciacus hirundinis, Hemiptera: Cimicidae). International Journal of Dermatology, 58: 115-116.

Response: We appreciate this point. To be consistent and given that only a single case exists for C. pipistrelli, we have removed it from the analysis for consistency.

Comment 3 (Facial involvement in infants, lines 340–341):

3 - About 'facial involvement", line 340-341, I have often observed this phenomenon, and I believe it is due to the fact that babies sleep in sleeping bags and toddlers are very often well covered when they sleep. Only the head is accessible for bed bugs.

Response: We thank the reviewer for this clinical insight. We have incorporated this explanation: that infants and toddler sleeping in bags or well-covered leave only the face accessible to bed bugs — into the Discussion. (Lines 299–300)

Comment 4 (Seasonal data):

4- Finally, it would have been interesting to consider, if available, seasonal data, especially for temperate climates, to see the distribution of bites throughout the year.

Response: We agree this would be valuable. We have added a note in the Limitations section acknowledging that seasonal information was rarely reported in the included case reports, and that future studies should aim to capture this variable.

---

## [Decision Letter · Decision Letter 1]

25 Mar 2026

Clinical Manifestations of Bed Bug Bites: A Systematic Review of Case Reports

PONE-D-25-69185R1

Dear Dr. Porras,

We’re pleased to inform you that your manuscript has been judged scientifically suitable for publication and will be formally accepted for publication once it meets all outstanding technical requirements.

Kind regards,

Jose Pietri

Academic Editor

PLOS One

Additional Editor Comments (optional):

Reviewers' comments:

Reviewer's Responses to Questions

**Comments to the Author**

Reviewer #1: All comments have been addressed

2. Is the manuscript technically sound, and do the data support the conclusions?

Reviewer #1: Yes

3. Has the statistical analysis been performed appropriately and rigorously?

Reviewer #1: Yes

4. Have the authors made all data underlying the findings in their manuscript fully available?

Reviewer #1: Yes

5. Is the manuscript presented in an intelligible fashion and written in standard English?

Reviewer #1: Yes

Reviewer #1: This revision adequately addresses the concerns raised in the first round of peer review.

There were just a couple of minor areas for consideration:

• The section beginning at line 185 still includes lengthy citation lists for each point; this reviewer isn’t certain that these citations need to be included here, but it is certainly not a deal-breaker.

• The REFERENCE section still shows inconsistency of capitalization (e.g. citation 19, 23, etc) and abbreviation/non-abbreviation of journal titles (e.g. citations 39, 42, etc.); perhaps this copy-editing will be done when typesetting the manuscript?

• The overall manuscript is still fairly lengthy, estimated at 4000+ words for the main body of the text. It would be more readable and thus effective if it were condensed to closer to 3000 words – there is some superfluous verbosity in the phrasing throughout, which could be reduced to reach this target. Once again, not essential but it would make for a more tidy paper.

.

Reviewer #1: No

---

## [Editor Report · Acceptance letter]

PONE-D-25-69185R1

PLOS One

Dear Dr. Porras-Villamil,

I'm pleased to inform you that your manuscript has been deemed suitable for publication in PLOS One. Congratulations! Your manuscript is now being handed over to our production team.

Kind regards,

on behalf of

Dr Jose Pietri

Academic Editor

PLOS One